# Stochastic Optimization of PCA with Capped MSG

**Raman Arora**
TTI-Chicago
Chicago, IL USA
arora@ttic.edu

**Andrew Cotter**
TTI-Chicago
Chicago, IL USA
cotter@ttic.edu

**Nathan Srebro**
Technion, Haifa, Israel
and TTI-Chicago
nati@ttic.edu

## Abstract

We study PCA as a stochastic optimization problem and propose a novel stochastic approximation algorithm which we refer to as "Matrix Stochastic Gradient" (MSG), as well as a practical variant, Capped MSG. We study the method both theoretically and empirically.

## 1 Introduction

Principal Component Analysis (PCA) is a ubiquitous tool used in many data analysis, machine learning and information retrieval applications. It is used to obtain a lower dimensional representation of a high dimensional signal that still captures as much of the original signal as possible. Such a low dimensional representation can be useful for reducing storage and computational costs, as complexity control in learning systems, or to aid in visualization.

PCA is typically phrased as a question about a fixed data set: given $n$ vectors in $\mathbb{R}^d$, what is the $k$-dimensional subspace that captures most of the variance in the data (or equivalently, that is best in reconstructing the vectors, minimizing the sum squared distances, or residuals, from the subspace)? It is well known that this subspace is the span of the leading $k$ components of the singular value decomposition of the data matrix (or equivalently of the empirical second moment matrix). Hence, the study of computational approaches for PCA has mostly focused on methods for finding the SVD (or leading components of the SVD) for a given $n \times d$ matrix (Oja & Karhunen, 1985; Sanger, 1989).

In this paper we approach PCA as a stochastic optimization problem, where the goal is to optimize a "population objective" based on i.i.d. draws from the population. In this setting, we have some unknown source ("population") distribution $\mathcal{D}$ over $\mathbb{R}^d$, and the goal is to find the $k$-dimensional subspace maximizing the (uncentered) variance of $\mathcal{D}$ inside the subspace (or equivalently, minimizing the average squared residual in the population), based on i.i.d. samples from $\mathcal{D}$. The main point here is that the true objective is not how well the subspace captures the *sample* (i.e. the "training error"), but rather how well the subspace captures the underlying source distribution (i.e. the "generalization error"). Furthermore, we are not concerned with capturing some "true" subspace, and so do not, for example, try to minimize the angle to such a subspace, but rather attempt to find a "good" subspace, i.e. one that is almost as good as the optimal one in terms of reconstruction error.

Of course, finding the subspace that best captures the sample is a very reasonable approach to PCA on the population. This is essentially an Empirical Risk Minimization (ERM) approach. However, when comparing it to alternative, perhaps computationally cheaper, approaches, we argue that one should not compare the error on the sample, but rather the population objective. Such a view can justify and favor computational approaches that are far from optimal on the sample, but are essentially as good as ERM *on the population*.

Such a population-based view of optimization has recently been advocated in machine learning, and has been used to argue for crude stochastic approximation approaches (online-type methods) over sophisticated deterministic optimization of the empirical (training) objective (i.e. "batch" methods) (Bottou & Bousquet, 2007; Shalev-Shwartz & Srebro, 2008). A similar argument was also

made in the context of stochastic optimization, where Nemirovski et al. (2009) argues for stochastic approximation (SA) approaches over ERM. approaches (a.k.a. ERM). Accordingly, SA approaches, mostly variants of Stochastic Gradient Descent, are often the methods of choice for many learning problems, especially when very large data sets are available (Shalev-Shwartz et al., 2007; Collins et al., 2008; Shalev-Shwartz & Tewari, 2009). We take the same view in order to advocate for, study, and develop stochastic approximation approaches for PCA.

In an empirical study of stochastic approximation methods for PCA, a heuristic "incremental" method showed very good empirical performance (Arora et al., 2012). However, no theoretical guarantees or justification were given for incremental PCA. In fact, it was shown that for some distributions it can converge to a suboptimal solution with high probability (see Section 5.2 for more about this "incremental" algorithm). Also relevant is careful theoretical work on online PCA by Warmuth & Kuzmin (2008), in which an online regret guarantee was established. Using an online-to-batch conversion, this online algorithm can be converted to a stochastic approximation algorithm with good iteration complexity, however the runtime for each iteration is essentially the same as that of ERM (i.e. of PCA on the sample), and thus senseless as a stochastic approximation method (see Section 3.3 for more on this algorithm).

In this paper we borrow from these two approaches and present a novel algorithm for stochastic PCA—the Matrix Stochastic Gradient (MSG) algorithm. MSG enjoys similar iteration complexity to Warmuth's and Kuzmin's algorithm, and in fact we present a unified view of both algorithms as different instantiations of Mirror Descent for the same convex relaxation of PCA. We then present the capped MSG algorithm, which is a more practical variant of MSG, has very similar updates to those of the "incremental" method, works well in practice, and does not get stuck like the "incremental" method. The Capped MSG algorithm is thus a clean, theoretically well founded method, with interesting connections to other stochastic/online PCA methods, and excellent practical performance—a "best of both worlds" algorithm.

## 2   Problem Setup

We consider PCA as the problem of finding the maximal (uncentered) variance $k$-dimensional subspace with respect to an (unknown) *distribution* $\mathcal{D}$ over $x \in \mathbb{R}^d$. We assume without loss of generality that the data are scaled in such a way that $\mathbb{E}_{x \sim \mathcal{D}}[\|x\|^2] \leq 1$. For our analysis, we also require that the fourth moment be bounded: $\mathbb{E}_{x \sim \mathcal{D}}[\|x\|^4] \leq 1$. We represent a $k$-dimensional subspace by an orthonormal basis, collected in the columns of a matrix $U$. With this parametrization, PCA is defined as the following stochastic optimization problem:

$$\text{maximize} : \mathbb{E}_{x \sim \mathcal{D}}[x^T U U^T x] \tag{2.1}$$
$$\text{subject to} : U \in \mathbb{R}^{d \times k}, U^T U = I.$$

In a stochastic optimization setting we do not have direct knowledge of the distribution $\mathcal{D}$, and instead may access it only through i.i.d. samples—these can be thought of as "training examples". As in other studies of stochastic approximation methods, we are less concerned with the number of required samples, and instead care mostly about the overall runtime required to obtain an $\epsilon$-suboptimal solution.

The standard approach to Problem 2.1 is empirical risk minimization (ERM): given samples $\{x_t\}_{t=1}^T$ drawn from $\mathcal{D}$, we compute the empirical covariance matrix $\hat{C} = \frac{1}{T} \sum_{t=1}^T x_t x_t^T$, and take the columns of $U$ to be the eigenvectors of $\hat{C}$ corresponding to the top-$k$ eigenvalues. This approach requires $O(d^2)$ memory and $O(d^2)$ operations just in order to compute the covariance matrix, plus some additional time for the SVD. We are interested in methods with much lower sample time and space complexity, preferably linear rather than quadratic in $d$.

## 3   MSG and MEG

A natural stochastic approximation (SA) approach to PCA is projected stochastic gradient descent (SGD) on Problem 2.1, with respect to $U$. This leads to the *stochastic power method*, for which, at each iteration, the following update is performed:

$$U^{(t+1)} = \mathcal{P}_{orth}\left(U^{(t)} + \eta x_t x_t^T\right) \tag{3.1}$$

Here, $x_t x_t^T$ is the gradient of the PCA objective w.r.t. $U$, $\eta$ is a step size, and $\mathcal{P}_{orth}(\cdot)$ projects its argument onto the set of matrices with orthonormal columns. Unfortunately, although SGD is well understood for convex problems, Problem 2.1 is non-convex. Consequently, obtaining a theoretical understanding of the stochastic power method, or of how the step size should be set, has proved elusive. Under some conditions, convergence to the optimal solution can be ensured, but no rate is known (Oja & Karhunen, 1985; Sanger, 1989; Arora et al., 2012).

Instead, we consider a re-parameterization of the PCA problem where the objective is convex. Instead of representing a linear subspace in terms of its basis matrix $U$, we parametrize it using the corresponding projection matrix $M = UU^T$. We can now reformulate the PCA problem as:

$$\text{maximize}: \ \mathbb{E}_{x \sim \mathcal{D}}[x^T M x] \tag{3.2}$$
$$\text{subject to}: \ M \in \mathbb{R}^{d \times d}, \lambda_i(M) \in \{0, 1\}, \text{rank}\, M = k$$

where $\lambda_i(M)$ is the $i^{th}$ eigenvalue of $M$.

We now have a convex (linear, in fact) objective, but the constraints are not convex. This prompts us relax the objective by taking the convex hull of the feasible region:

$$\text{maximize}: \ \mathbb{E}_{x \sim \mathcal{D}}[x^T M x] \tag{3.3}$$
$$\text{subject to}: \ M \in \mathbb{R}^{d \times d}, 0 \preceq M \preceq I, \text{tr}\, M = k$$

Since the objective is linear, and the feasible regiuon is the convex hull of that of Problem 3.2, an optimal solution is always attained at a "vertex", i.e. a point on the boundary of the original constraints. The optima of the two objectives are thus the same (strictly speaking—every optimum of Problem 3.2 is also an optimum of Problem 3.3), and solving Problem 3.3 is equivalent to solving Problem 3.2.

Furthermore, if a suboptimal solution for Problem 3.3 is not rank-$k$, i.e. is not a feasible point of Problem 3.2, we can easily sample from it to obtain a rank-$k$ solution with the same objective function value (in expectation). This is shown by the following result of Warmuth & Kuzmin (2008):

**Lemma 3.1** (Rounding (Warmuth & Kuzmin, 2008)). *Any feasible solution of Problem 3.3 can be expressed as a convex combination of at most $d$ feasible solutions of Problem 3.2.*

Algorithm 2 of Warmuth & Kuzmin (2008) shows how to efficiently find such a convex combination. Since the objective is linear, treating the coefficients of the convex combination as defining a discrete distribution, and sampling according to this distribution, yields a rank-$k$ matrix with the desired expected objective function value.

### 3.1 Matrix Stochastic Gradient

Performing SGD on Problem 3.3 (w.r.t. the variable $M$) yields the following update rule:

$$M^{(t+1)} = \mathcal{P}\left(M^{(t)} + \eta x_t x_t^T\right), \tag{3.4}$$

The projection is now performed onto the (convex) constraints of Problem 3.3. This gives the **Matrix Stochastic Gradient (MSG)** algorithm, which, in detail, consists of the following steps:

1. Choose a step-size $\eta$, iteration count $T$, and starting point $M^{(0)}$.
2. Iterate the update rule (Equation 3.4) $T$ times, each time using an independent sample $x_t \sim \mathcal{D}$.
3. Average the iterates as $\bar{M} = \frac{1}{T} \sum_{t=1}^{T} M^{(t)}$.
4. Sample a rank-$k$ solution $\tilde{M}$ from $\bar{M}$ using the rounding procedure discussed in the previous section.

Analyzing MSG is straightforward using a standard SGD analysis:

**Theorem 1.** *After $T$ iterations of MSG (on Problem 3.3), with step size $\eta = \sqrt{\frac{k}{T}}$, and starting at $M^{(0)} = 0$,*

$$\mathbb{E}[\mathbb{E}_{x \sim \mathcal{D}}[x^T \tilde{M} x]] \geq \mathbb{E}_{x \sim \mathcal{D}}[x^T M^* x] - \frac{1}{2}\sqrt{\frac{k}{T}},$$

*where the expectation is w.r.t. the i.i.d. samples $x_1, \ldots, x_T \sim \mathcal{D}$ and the rounding, and $M^*$ is the optimum of Problem 3.2.*

---

**Algorithm 1** Matrix stochastic gradient (MSG) update: compute an eigendecomposition of $M'+\eta xx^T$ from a rank-$m$ eigendecomposition $M'=U'\text{diag}(\sigma')(U')^T$ and project the resulting solution onto the constraint set. The computational cost is dominated by the matrix multiplication on lines 4 or 7 costing $O(m^2d)$ operations.

```
     msg-step (d, k, m : N, U' : R^{d×m}, σ' : R^m, x : R^d, η : R)
1       x̂ ← √η(U')^T x; x_⊥ ← √η x − U'x̂; r ← ‖x_⊥‖;
2       if r > 0
3           V, σ ← eig([diag(σ') + x̂x̂^T, rx̂; rx̂^T, r^2]);
4           U ← [U', x_⊥/r]V;
5       else
6           V, σ ← eig(diag(σ') + x̂x̂^T);
7           U ← U'V;
8       σ ← distinct eigenvalues in σ; κ ← corresponding multiplicities;
9       σ ← project (d, k, m, σ, κ);
10      return U, σ;
```

---

*Proof.* The SGD analysis of Nemirovski & Yudin (1983) yields that:

$$\mathbb{E}[x^T M^* x - x^T \bar{M} x] \leq \frac{\eta}{2}\mathbb{E}_{x\sim\mathcal{D}}[\|g\|_F^2] + \frac{\|M^* - M^{(0)}\|_F^2}{2\eta T} \tag{3.5}$$

where $g = xx^T$ is the gradient of the PCA objective. Now, $\mathbb{E}_{x\sim\mathcal{D}}[\|g\|_F^2] = \mathbb{E}_{x\sim\mathcal{D}}[\|x\|^4] \leq 1$ and $\left\|M^* - M^{(0)}\right\|_F^2 = \|M^*\|_F^2 = k$. In the last inequality, we used the fact that $M^*$ has $k$ eigenvalues of value 1 each, and hence $\|M^*\|_F = \sqrt{k}$. □

### 3.2 Efficient Implementation and Projection

A naïve implementation of the MSG update requires $O(d^2)$ memory and $O(d^2)$ operations per iteration. In this section, we show how to perform this update efficiently by maintaining an up-to-date eigendecomposition of $M^{(t)}$. Pseudo-code for the update may be found in Algorithm 1. Consider the eigendecomposition $M^{(t)} = U'\text{diag}(\sigma)(U')^T$ at the $t^{th}$ iteration, where $\text{rank}(M^{(t)}) = k_t$ and $U' \in \mathbb{R}^{d\times k_t}$. Given a new observation $x_t$, the eigendecomposition of $M^{(t)} + \eta x_t x_t^T$ can be updated efficiently using a $(k_t+1)\times(k_t+1)$ SVD (Brand, 2002; Arora et al., 2012) (steps 1-7 of Algorithm 1). This rank-one eigen-update is followed by projection onto the constraints of Problem 3.3, invoked as `project` in step 8 of Algorithm 1, discussed in the following paragraphs and given as Algorithm 2. The projection procedure is based on the following lemma[1]. See supplementary material for a proof.

**Lemma 3.2.** *Let $M' \in \mathbb{R}^{d\times d}$ be a symmetric matrix, with eigenvalues $\sigma'_1, \ldots, \sigma'_d$ and associated eigenvectors $v'_1, \ldots, v'_d$. Its projection $M = \mathcal{P}(M')$ onto the feasible region of Problem 3.3 with respect to the Frobenius norm, is the unique feasible matrix which has the same eigenvectors as $M'$, with the associated eigenvalues $\sigma_1, \ldots, \sigma_d$ satisfying:*

$$\sigma_i = \max\left(0, \min\left(1, \sigma'_i + S\right)\right)$$

*with $S \in \mathbb{R}$ being chosen in such a way that $\sum_{i=1}^d \sigma_i = k$.*

This result shows that projecting onto the feasible region amounts to finding the value of $S$ such that, after shifting the eigenvalues by $S$ and clipping the results to $[0, 1]$, the result is feasible. Importantly, the projection operates *only* on the eigenvalues. Algorithm 2 contains pseudocode which finds $S$ from a list of eigenvalues. It is optimized to efficiently handle repeated eigenvalues—rather than receiving the eigenvalues in a length-$d$ list, it instead receives a length-$n$ list containing only the *distinct* eigenvalues, with $\kappa$ containing the corresponding multiplicities. In Sections 4 and 5, we will see why this is an important optimization. The central idea motivating the algorithm is that, in a sorted array of eigenvalues, all elements with indices below some threshold $i$ will be clipped to 0, and all of those with indices above another threshold $j$ will be clipped to 1. The pseudocode simply searches over all possible pairs of such thresholds until it finds the one that works.

The rank-one eigen-update combined with the fast projection step yields an efficient MSG update that requires $O(dk_t)$ memory and $O(dk_t^2)$ operations per iteration (recall that $k_t$ is the rank of the

**Algorithm 2** Routine which finds the $S$ of Lemma 3.2. It takes as parameters the dimension $d$, "target" subspace dimension $k$, and the number of *distinct* eigenvalues $n$ of the current iterate. The length-$n$ arrays $\sigma'$ and $\kappa'$ contain the distinct eigenvalues and their multiplicities, respectively, of $M'$ (with $\sum_{i=1}^{n} \kappa'_i = d$). Line 1 sorts $\sigma'$ and re-orders $\kappa'$ so as to match this sorting. The loop will be run at most $2n$ times (once for each possible increment to $i$ or $j$ on lines 12–15), so the computational cost is dominated by that of the sort: $O(n \log n)$.

```
    project (d, k, n : ℕ, σ' : ℝⁿ, κ' : ℕⁿ)
1       σ', κ' ← sort(σ', κ');
2       i ← 1; j ← 1; sᵢ ← 0; sⱼ ← 0; cᵢ ← 0; cⱼ ← 0;
3       while  i ≤ n
4           if  (i < j)
5               S ← (k − (sⱼ − sᵢ) − (d − cⱼ))/(cⱼ − cᵢ);
6               b ← (
7                   (σ'ᵢ + S ≥ 0) and (σ'ⱼ₋₁ + S ≤ 1)
8                   and ((i ≤ 1) or (σ'ᵢ₋₁ + S ≤ 0))
9                   and ((j ≥ n) or (σ'ⱼ₊₁ ≥ 1))
10              );
11              return S if b;
12          if  (j ≤ n) and (σ'ⱼ − σ'ᵢ ≤ 1)
13              sⱼ ← sⱼ + κ'ⱼσ'ⱼ; cⱼ ← cⱼ + κ'ⱼ; j ← j + 1;
14          else
15              sᵢ ← sᵢ + κ'ᵢσ'ᵢ; cᵢ ← cᵢ + κ'ᵢ; i ← i + 1;
16      return error;
```

iterate $M^{(t)}$). This is a significant improvement over the $O(d^2)$ memory and $O(d^2)$ computation required by a standard implementation of MSG, if the iterates have relatively low rank.

### 3.3   Matrix Exponentiated Gradient

Since $M$ is constrained by its trace, and not by its Frobenius norm, it is tempting to consider mirror descent (MD) (Beck & Teboulle, 2003) instead of SGD updates for solving Problem 3.3. Recall that Mirror Descent depends on a choice of "potential function" $\Psi(\cdot)$ which should be chosen according to the geometry of the feasible set *and* the subgradients (Srebro et al., 2011). Using the squared Frobenius norm as a potential function, i.e. $\Psi(M) = \|M\|_F^2$, yields SGD, i.e. the MSG updates Equation 3.4. The trace-norm constraint suggests using the von Neumann entropy as the potential function, i.e. $\Psi_h(M) = \sum_i \lambda_i(M) \log \lambda_i(M)$. This leads to multiplicative updates, yielding what we refer to as the Matrix Exponentiated Gradient (MEG) algorithm, which is similar to that of (Warmuth & Kuzmin, 2008). In fact, Warmuth and Kuzmin's algorithm exactly corresponds to online Mirror Descent on Problem 3.3 with this potential function, but takes the optimization variable to be $M_\perp = I - M$ (with the constraints $\operatorname{tr} M_\perp = d - k$ and $0 \preceq M_\perp \preceq I$). In either case, using the entropy potential, despite being well suited for the trace-geometry, does not actually lead to a better dependence[2] on $d$ or $k$, and a Mirror Descent-based analysis again yields an excess loss of $\sqrt{k/T}$. Warmuth and Kuzmin present an "optimistic" analysis, with a dependence on the "reconstruction error" $L^* = \mathbb{E}[x^T(I - M^*)x]$, which yields an excess error of $O\left(\sqrt{\frac{L^* k \log(d/k)}{T}} + \frac{k \log(d/k)}{T}\right)$ (their logarithmic term can be avoided by a more careful analysis).

## 4   MSG runtime and the rank of the iterates

As we saw in Sections 3.1 and 3.2, MSG requires $O(k/\epsilon^2)$ iterations to obtain an $\epsilon$-suboptimal solution, and each iteration costs $O(k_t^2 d)$ operations, where $k_t$ is the rank of iterate $M^{(t)}$. This yields a total runtime of $O(\bar{k}^2 dk/\epsilon^2)$, where $\bar{k}^2 = \sum_{t=1}^{T} k_t^2$. Clearly, the runtime for MSG depends critically on the rank of the iterates. If $k_t$ is as large as $d$, then MSG achieves a runtime that is cubic in the dimensionality. On the other hand, if the rank of the iterates is $O(k)$, the runtime is linear in the dimensionality. Fortunately, in practice, each $k_t$ is typically much lower than $d$. The reason for this is that the MSG update performs a rank-1 update followed by a projection onto the constraints. Since $M' = M^{(t)} + \eta x_t x_t^T$ will have a *larger* trace than $M^{(t)}$ (i.e. $\operatorname{tr} M' \geq k$), the projection, as is

shown by Lemma 3.2, will *subtract* a quantity $S$ from every eigenvalue of $M'$, clipping each to $0$ if it becomes negative. Therefore, each MSG update will increase the rank of the iterate by at most 1, and has the potential to decrease it, perhaps significantly. It's very difficult to theoretically quantify how the rank of the iterates will evolve over time, but we have observed empirically that the iterates do tend to have relatively low rank.

We explore this issue in greater detail experimentally, on a distribution which we expect to be difficult for MSG. To this end, we generated data from known 32-dimensional distributions with diagonal covariance matrices $\Sigma = \text{diag}(\sigma/\|\sigma\|)$, where $\sigma_i = \tau^{-i}/\sum_{j=1}^{32}\tau^{-j}$, for $i = 1,\ldots,32$ and for some $\tau > 1$. Observe that $\Sigma^{(k)}$ has a smoothly-decaying set of eigenvalues and the rate of decay is controlled by $\tau$. As $\tau \to 1$, the spectrum becomes flatter resulting in distributions that present challenging test cases for MSG. We experimented with $\tau = 1.1$ and $k \in \{1, 2, 4\}$, where $k$ is the desired subspace dimension used by each algorithm. The data is generated by sampling the $i^{th}$ standard unit basis vector $e_i$ with probability $\sqrt{\Sigma_{ii}}$. We refer to this as the "orthogonal distribution", since it is a discrete distribution over 32 orthogonal vectors.

In Figure 1, we show the results with $k = 4$. We can see from the left-hand plot that MSG maintains a subspace of dimension around 15. The plot on the right shows how the set of nonzero eigenvalues of the MSG iterates evolves over time, from which we can see that many of the extra dimensions are "wasted" on very small eigenvalues, corresponding to directions which leave the state matrix only a handful of iterations after they enter. This suggests that constraining $k_t$ can lead to significant speedups and motivates capped MSG updates discussed in the next section.

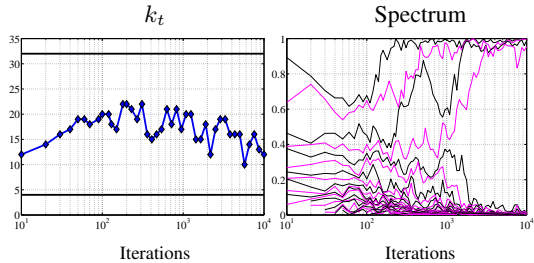

Figure 1: The ranks $k_t$ (left) and the eigenvalues (right) of the MSG iterates $M^{(t)}$.

## 5 Capped MSG

While, as was observed in the previous section, MSG's iterates will tend to have ranks $k_t$ smaller than $d$, they will nevertheless also be larger than $k$. For this reason, we recommend imposing a hard constraint $K$ on the rank of the iterates:

$$\text{maximize}: \ \mathbb{E}_{x\sim\mathcal{D}}[x^T M x] \qquad (5.1)$$
$$\text{subject to}: \ M \in \mathbb{R}^{d\times d}, 0 \preceq M \preceq I$$
$$\text{tr}\,M = k, \text{rank}\,M \leq K$$

We will refer to MSG where the projection is replaced with a projection onto the constraints of Problem 5.1 (i.e. where the iterates are SGD iterates on Problem 5.1) as "capped MSG". As before, as long as $K \geq k$, Problem 5.1 and Problem 3.3 have the same optimum, it is achieved at a rank-$k$ matrix, and the extra rank constraint in Problem 5.1 is inactive at the optimum. However, the rank constraint does affect the iterates, especially since Problem 5.1 is no longer convex. Nonetheless if $K > k$ (i.e. the hard rank-constraint $K$ is *strictly* larger than the target rank $k$), then we can easily check if we are at a global optimum of Problem 5.1, and hence of Problem 3.3: if the capped MSG algorithm converges to a solution of rank $K$, then the upper bound $K$ should be increased. Conversely, if it has converged to a rank-deficient solution, then it must be the global optimum. There is thus an advantage in using $K > k$, and we recommend setting $K = k + 1$, as we do in our experiments, and increasing $K$ only if a rank deficient solution is not found in a timely manner.

If we take $K = k$, then the only way to satisfy the trace constraint is to have all non-zero eigenvalues equal to one, and Problem 5.1 becomes identical to Problem 3.2. The detour through the convex objective of Problem 3.3 allows us to increase the search rank $K$, allowing for more flexibility in the iterates, while still forcing each iterate to be low-rank, and each update to therefore be efficient, through the rank constraint.

### 5.1 Implementing the projection

The only difference between the implementation of MSG and capped MSG is in the projection step. Similar reasoning to that which was used in the proof of Lemma 3.2 shows that if $M^{(t+1)} = \mathcal{P}(M')$

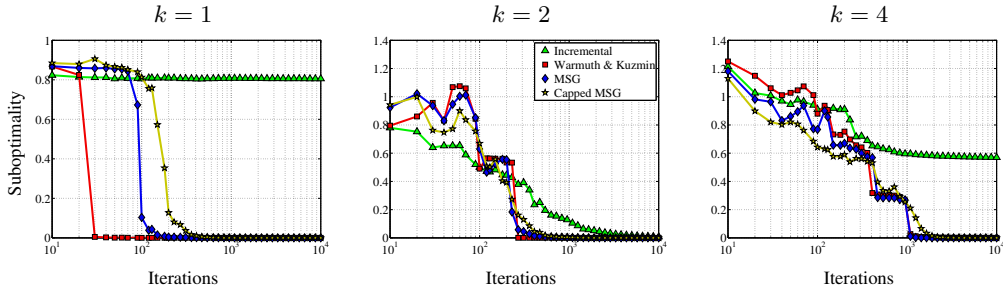

Figure 2: Comparison on simulated data for different values of parameter $k$.

with $M' = M^{(t)} + \eta x_t x_t^T$, then $M^{(t)}$ and $M'$ are simultaneously diagonalizable, and therefore we can consider only how the projection acts on the eigenvalues. Hence, if we let $\sigma'$ be the vector of the eigenvalues of $M'$, and suppose that more than $K$ of them are nonzero, then there will be a a size-$K$ subset of $\sigma'$ such that applying Algorithm 2 to this set gives the projected eigenvalues. Since we perform only a rank-1 update at every iteration, we must check at most $K$ possibilities, at a total cost of $O(K^2 \log K)$ operations, which has no effect on the asymptotic runtime because Algorithm 1 requires $O(K^2 d)$ operations.

## 5.2   Relationship to the incremental PCA method

The capped MSG algorithm with $K = k$ is similar to the incremental algorithm of Arora et al. (2012), which maintains a rank-$k$ approximation of the covariance matrix and updates according to:

$$M^{(t+1)} = \mathcal{P}_{\text{rank-}k}\left(M^{(t)} + x_t x_t^T\right)$$

where the projection is onto the set of rank-$k$ matrices. Unlike MSG, the incremental algorithm does not have a step-size. Updates can be performed efficiently by maintaining an eigendecomposition of each iterate, just as was done for MSG (see Section 3.2).

In a recent survey of stochastic algorithms for PCA (Arora et al., 2012), the incremental algorithm was found to perform extremely well in practice–it was the best, in fact, among the compared algorithms. However, there exist cases in which it can get stuck at a suboptimal solution. For example, If the data are drawn from a discrete distribution $\mathcal{D}$ which samples $[\sqrt{3}, 0]^T$ with probability $1/3$ and $[0, \sqrt{2}]^T$ with probability $2/3$, and one runs the incremental algorithm with $k = 1$, then it will converge to $[1, 0]^T$ with probability $5/9$, despite the fact that the maximal eigenvector is $[0, 1]^T$. The reason for this failure is essentially that the orthogonality of the data interacts poorly with the low-rank projection: any update which does not entirely displace the maximal eigenvector in one iteration will be removed entirely by the projection, causing the algorithm to fail to make progress. The capped MSG algorithm with $K > k$ will not get stuck in such situations, since it will use the additional dimensions to search in the new direction. Only as it becomes more confident in its current candidate will the trace of $M$ become increasingly concentrated on the top $k$ directions. To illustrate this empirically, we generalized this example by generating data using the 32-dimensional "orthogonal" distribution described in Section 4. This distribution presents a challenging test-case for MSG, capped MSG and the incremental algorithm. Figure 2 shows plots of individual runs of MSG, capped MSG with $K = k + 1$, the incremental algorithm, and Warmuth and Kuzmin's algorithm, all based on the same sequence of samples drawn from the orthogonal distribution. We compare algorithms in terms of the suboptimality on the population objective based on the largest $k$ eigenvalues of the state matrix $M^{(t)}$. The plots show the incremental algorithm getting stuck for $k \in \{1, 4\}$, and the others intermittently plateauing at intermediate solutions before beginning to again converge rapidly towards the optimum. This behavior is to be expected on the capped MSG algorithm, due to the fact that the dimension of the subspace stored at each iterate is constrained. However, it is somewhat surprising that MSG and Warmuth and Kuzmin's algorithm behaved similarly, and barely faster than capped MSG.

## 6   Experiments

We also compared the algorithms on the real-world MNIST dataset, which consists of $70,000$ binary images of handwritten digits of size $28 \times 28$, resulting in a dimensionality of $784$. We pre-normalized the data by mean centering the feature vectors and scaling each feature by the product of its standard

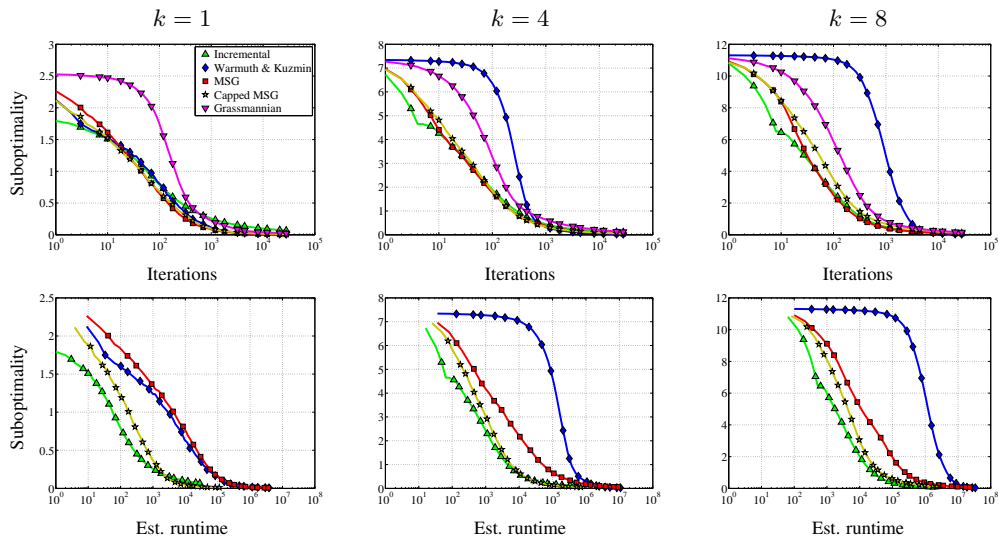

Figure 3: Comparison on the MNIST dataset. The top row of plots shows suboptimality as a function of iteration count, while the bottom row suboptimality as a function of estimated runtime $\sum_{s=1}^{t}(k'_s)^2$.

deviation and the data dimension, so that each feature vector is zero mean and unit norm in expectation. In addition to MSG, capped MSG, the incremental algorithm and Warmuth and Kuzmin's algorithm, we also compare to a Grassmannian SGD algorithm (Balzano et al., 2010). All algorithms except the incremental algorithm have a step-size parameter. In these experiments, we ran each algorithm with decreasing step sizes $\eta_t = c/\sqrt{t}$ for $c \in \{2^{-12}, 2^{-11}, \ldots, 2^5\}$ and picked the best $c$, in terms of the average suboptimality over the run, on a validation set. Since we cannot evaluate the true population objective, we estimate it by evaluating on a held-out test set. We use 40% of samples in the dataset for training, 20% for validation (tuning step-size), and 40% for testing. We are interested in learning a maximum variance subspace of dimension $k \in \{1, 4, 8\}$ in a single "pass" over the training sample. In order to compare MSG, capped MSG, the incremental algorithm and Warmuth and Kuzmin's algorithm in terms of runtime, we calculate the dominant term in the computational complexity: $\sum_{s=1}^{t}(k'_s)^2$. The results are averaged over 100 random splits into train-validation-test sets.

We can see from Figure 3 that the incremental algorithm makes the most progress per iteration and is also the fastest of all algorithms. MSG is comparable to the incremental algorithm in terms of the the progress made per iteration. However, its runtime is slightly worse because it will often keep a slightly larger representation (of dimension $k_t$). The capped MSG variant (with $K = k + 1$) is significantly faster–almost as fast as the incremental algorithm, while, as we saw in the previous section, being less prone to getting stuck. Warmuth and Kuzmin's algorithm fares well with $k = 1$, but its performance drops for higher $k$. Inspection of the underlying data shows that, in the $k \in \{4, 8\}$ experiments, it also tends to have a larger $k_t$ than MSG in these experiments, and therefore has a higher cost-per-iteration. Grassmannian SGD performs better than Warmuth and Kuzmin's algorithm, but much worse than MSG and capped MSG.

## 7   Conclusions

In this paper, we presented a careful development and analysis of MSG, a stochastic approximation algorithm for PCA, which enjoys good theoretical guarantees and offers a computationally efficient variant, capped MSG. We show that capped MSG is well-motivated theoretically and that it does not get stuck at a suboptimal solution. Capped MSG is also shown to have excellent empirical performance and it therefore is a much better alternative to the recently proposed incremental PCA algorithm of Arora et al. (2012). Furthermore, we provided a cleaner interpretation of PCA updates of Warmuth & Kuzmin (2008) in terms of Matrix Exponentiated Gradient (MEG) updates and showed that both MSG and MEG can be interpreted as mirror descent algorithms on the same relaxation of the PCA optimization problem but with different distance generating functions.

## Footnotes

[1]Our projection problem onto the capped simplex, even when seen in the vector setting, is substantially different from Duchi et al. (2008). We project onto the set $\{0 \leq \sigma \leq 1, \|\sigma\|_1 = k\}$ in Problem 3.3 and $\{0 \leq \sigma \leq 1, \|\sigma\|_1 = k, \|\sigma\|_0 \leq K\}$ in Problem 5.1 whereas Duchi et al. (2008) project onto $\{0 \leq \sigma, \|\sigma\|_1 = k\}$.

[2]This is because in our case, due to the other constraints, $\|M^*\|_F = \sqrt{\operatorname{tr} M^*}$. Furthermore, the SGD analysis depends on the Frobenius norm of the stochastic gradients, but since all stochastic gradients are rank one, this is the same as their spectral norm, which comes up in the entropy-case analysis, and again there is no benefit.

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
