[Supplementary Material · NIPS.2013.Capped.MSG.Appendix.pdf]

# A  Proof of Lemma 3.2

**Lemma 3.2.** *Let $M' \in \mathbb{R}^{d \times d}$ be a symmetric matrix, with eigenvalues $\sigma'_1, \ldots, \sigma'_d$ and associated eigenvectors $v'_1, \ldots, v'_d$. If $M = \mathcal{P}(M')$ projects $M'$ onto the feasible region of Problem 3.3 with respect to the Frobenius norm, then $M$ will be the unique feasible matrix which has the same set of eigenvectors as $M'$, with the associated eigenvalues $\sigma_1, \ldots, \sigma_d$ satisfying:*

$$\sigma_i = \max\left(0, \min\left(1, \sigma'_i + S\right)\right)$$

*with $S \in \mathbb{R}$ being chosen in such a way that $\displaystyle\sum_{i=1}^{d} \sigma_i = k$.*

*Proof.* The problem of finding $M$ can be written in the form of a convex optimization problem as:

$$\begin{aligned}\text{minimize} &: \ \|M - M'\|_F^2 \\ \text{subject to} &: \ 0 \preceq M \preceq I, \operatorname{tr} M = k.\end{aligned}$$

Because the objective is strongly convex, and the constraints are convex, this problem must have a unique solution. Letting $\sigma_1, \ldots, \sigma_d$ and $v_1, \ldots, v_d$ be the eigenvalues and associated eigenvectors of $M$, we may write the KKT first-order optimality conditions (Boyd & Vandenberghe, 2004) as:

$$0 = M - M' + \mu I - \sum_{i=1}^{d} \alpha_i v_i v_i^T + \sum_{i=1}^{d} \beta_i v_i v_i^T, \tag{A.1}$$

where $\mu$ is the Lagrange multiplier for the constraint $\operatorname{tr} M = k$, and $\alpha_i, \beta_i \geq 0$ are the Lagrange multipliers for the constraints $0 \preceq M$ and $M \preceq I$, respectively. The complementary slackness conditions are that $\alpha_i \sigma_i = \beta_i (\sigma_i - 1) = 0$. In addition, $M$ must be feasible.

Because every term in Equation A.1 *except* for $M'$ has the same set of eigenvectors as $M$, it follows that an optimal $M$ must have the same set of eigenvectors as $M'$, so we may take $v_i = v'_i$, and write Equation A.1 purely in terms of the eigenvalues:

$$\sigma_i = \sigma'_i - \mu + \alpha_i - \beta_i.$$

Complementary slackness and feasibility with respect to the constraints $0 \preceq M \preceq I$ gives that if $0 \leq \sigma'_i - \mu \leq 1$, then $\sigma_i = \sigma'_i - \mu$. Otherwise, $\alpha_i$ and $\beta_i$ will be chosen so as to clip $\sigma_i$ to the active constraint:

$$\sigma_i = \max\left(0, \min\left(1, \sigma'_i - \mu\right)\right).$$

Primal feasibility with respect to the constraint $\operatorname{tr} M = k$ gives that $\mu$ must be chosen in such a way that $\operatorname{tr} M = k$, completing the proof. $\qquad\square$