[Reviews · NeurIPS 2013]

Submitted by Assigned_Reviewer_7

This paper explores online stochastic approximation algorithms for PCA. The authors
provide a few apparently novel suggestions and heuristics that extend existing
methods. However, the main contribution here is really about unifying and elucidating the disparate perspectives taken in the literature, and analyzing common approaches to online PCA. The paper is careful, well-written and clear, however it is as if the authors had four or five smaller, somewhat unrelated results they decided to tie together under the rather broad banner of 'Matrix stochastic gradient for PCA' and related concepts. As such, the discussion could benefit from further effort devoted to connecting the various pieces into a coherent whole.

Having said this, the points made in this paper, even if disjoint at times, are important ones, and moreover can be hard to piece together from the literature without significant effort. The authors have absorbed some (but by no means all -- see comments below) of the relevant work, and have pointed out commonalities, differences, subtleties, and a few low-hanging improvements. This, in my opinion, is valuable enough to make up for the paper's shortcomings. As noted above, the authors are careful, and provide a bit of theory in a language and setting that is probably much easier to digest compared to what may be found in the references (which
may be comprehensive, but require a greater time investment to get the "gist" of the proofs and the behavior of the algorithms).

Other comments:
- In the introduction it is implied that taking a population approach is something of fundamental innovation offered by the paper. That is a stretch. In the end, we're
still talking about online PCA, and calling it by another name doesn't appear to offer any concrete, substantive additions to the conversation.
- The abstract claims that Eq (3.3) is "novel", however working with the projection
matrix rather than the matrix of eigenvectors is common, and may even be viewed as
an alternative representation of the Grassmann manifold. This equation, and others,
probably overlap quite substantially with the approaches taken in e.g.
+ "Optimization Algorithms on Matrix Manifolds" by Absil, P., Mahony, R., Sepulchre, R. ( this reference also addresses more general gradient flows, stepsize, retraction ("projection"), and convergence issues).
+ S. Bonnabel, "Stochastic gradient descent on Riemannian manifolds" (arXiv)
+ A. Edelman et al. "THE GEOMETRY OF ALGORITHMS WITH ORTHOGONALITY CONSTRAINTS"
- The perspective of gradient descent on matrix manifolds is, to this reviewer, a major omission. The paper could greatly benefit from a discussion as to how this slice of the literature fits in to the picture, in terms of both practical algorithms and theory. I understand that space is a constraint. But if the authors decide to ultimately write a journal version of the submission, careful, considered inclusion of the above would make for a fantastic, authoritative paper bridging all of the major points of view on the topic.
- In light of the above references, a "theoretical understanding" of the power method may not be so elusive after all! (statement at line 100)
- Can the projection step also be derived using standard tools from proximal methods?
- The experiments are well chosen to reveal the behavior of the algorithms in question.
- If the principal components are needed after all is said and done, do you include anywhere the cost of computing the factorization M=UU'?
Summary: Cons: The paper's continuity could be improved; claims of novelty could be softened; matrix manifold methods should be included in the picture, as the related literature overlaps substantially but has been ignored. Pros: A clear, unifying perspective on (online) matrix stochastic gradient descent for the eigenvalue/vector problem that contributes meaningfully to a gap in the literature.

Submitted by Assigned_Reviewer_8

The paper proposes an online-type method for PCA, or more precisely for finding the best k-dimensional subspace maximizing the variance with the true test distribution (as opposed to classical PCA on the training sample).

The technique builds upon Warmuth & Kuzmin (2008). On the theory side, the paper gives interesting new insights that might be useful for other related methods as well. On the experiments side, the new method has some advantages over the existing ones are in some regimes. I consider the value of the contribution rather on the theory side.

*** Theory
The most important novel contribution in my opinion is the explicit running time analysis of O(k'^2dk/ε^2), for k'^2 being the sum of the squared ranks of the iterates.
Furthermore, the interpretation of both the new algorithm and (Warmuth & Kuzmin, 2008) as convex online methods on (3.2) is definitely valuable.

*** Experiments
The performed experiments seem relatively convincing. The proposed algorithm is practically not better than the incremental one by (Arora et al., 2012), but has stronger theoretical guarantees, which I think is important.
Questions:
a) Apparently the important comparison on the running time was not on the real implementations, but only stating the theoretical time bounds of \sum_s (k'_s )^2.
What is the reason for this, what is the deviation of the different methods from this time, and how does the real running time look like?

b) Also, in practice one is usually allowed a few passes over the data (vs. just one). It would be important to comment about the simpler approach of just running the methods for k=1, and then continue with the residuals and do the next pass. Are there any new insights as compared to the existing work in this setting?

c) Why do the k=1 results look so different in Figure 1 as compared to Figure 2?

* The clarity of the writeup could be improved, many parts are very dense currently due to the page constraints.

*** Minor issues
-Lemma 3.1: If only existence is needed, then Caratheodory alone already gives us the result, right? (need to state what the true dimension is though)
-l068: "senseless" is a bit strong when thinking that the iteration complexity is the same and as it is later shown that it's online mirror descent
-l075: _C_apped
-l137: Algorithm 4.1 is their algo 4 ?
-mention that (2.1) is non-convex when it's defined
-explain why it becomes the power method in a special case
-l317: constraint k<=K "inactive": more precisely, also for k=K?

*** Update:
I considered the author feedback, and hope the authors will implement the discussed issues in the final version
Summary: The paper proposes a new online-type method for PCA, or more precisely for finding the best k-dimensional subspace maximizing the variance with the true test distribution (as opposed to classical PCA on the training sample).

Submitted by Assigned_Reviewer_9

Principal Component Analysis (PCA) is a basic statistical subroutine that is used in many applications. The goal is to find the subspace that maximizes the projected variance. The authors consider a related objective, that seems interesting and seems to depart from much of the existing literature. The viewpoint in this paper is that the observed data comes from some underlying distribution, and we would like to find a subspace that has large projected variance for that distribution (not necessarily for the observed data, although those objectives should not be that different). In particular, the goal is not to recover the true subspace up to some principal angle, but to have the subspace found by the algorithm be almost as good as the best in terms of capturing the variance of the underlying distribution.

The authors propose an algorithm that they call the Matrix Stochastic Gradient (MSG) algorithm, which is closely related to an algorithm of Warmuth and Kuzmin. The authors give a cohesive presentation of both algorithms in terms of Mirror Descent, and use this intuition to propose a capped variation of their algorithm. The authors establish a few interesting theoretical properties of their algorithm, and also show empirical justification too.
Summary: I think this paper is interesting, and would be a good fit at NIPS but I am not an expert in existing work, so cannot say with confidence that this paper would also be interesting to experts.
Author Feedback

Author rebuttal: We thank the reviewers for their helpful comments. We found the reviews fair and overall accurate---there are no significant errors or misunderstandings in the reviews that need to be corrected. Here are some responses to specific questions raised:

Reviewer 1:

Thanks for the references. We do refer to and compare with Grassmannian SGD (GROUSE), which is a gradient descent algorithm on a matrix manifold, on line 403, in Sec 6.

Equation 3.3: although the projection matrix is indeed just an alternative representation for a point on the Grassmannian manifold, the difference lies in that gradients are calculated w.r.t. this matrix, which is different than the gradient on the Grassmannian manifold.

MSG method can be derived as a proximal methods if we consider the same convex relaxation as we studied in this paper, and take as the potential function the squared Frobenius norm of the projection matrix--this is essentially the path taken in the paper, since gradient descent can be derived as a proximal method w.r.t. the square Euclidean norm potential function.

There is no additional cost of computing eigenvectors (M=UU’) because our efficient implementation (Sec. 3.2) maintains the eigenvectors (U) at each iteration instead of the projection matrix (M) itself.

Reviewer 2:

Measuring “steps” versus runtime: we find counting vector operations a more meaningful and reproducible measure than actual runtime, since runtime is very implementation and architecture dependent. We are not comparing here different packages but rather different methods/approaches and so would rather not be implementation-dependent, so that we don’t run the risk of saying some method is worse just because it was not implemented well. That said, we saw very similar trends in the actual runtime plots.

Incrementally building the subspace using multiple passes over the data: This could indeed be an interesting comparison that we did not do. We expect the total runtime with multiple passes to be higher, since in pass k+1, after we already have a rank-k subspace, we have to keep projecting out the k subspace we already have, which is similar in runtime to just optimizing over a rank-k subspace, as we do in our approach.

The k=1 results look different in Figures 2 and 3 because Figure 2 shows the results of experiments on a simulated data distribution which was constructed to be particularly challenging for the incremental and capped MSG algorithms, while figure 3 shows results on the (much easier) MNIST dataset. In particular, the incremental algorithm got "stuck" for k=1 in Figure 2, while, thanks to the additional dimensions, it managed to (eventually) converge for larger k.

“Algorithm 4.1 of Warmuth & Kuzmin (2008)” should have been “Algorithm 2 of Warmuth & Kuzmin (2008)”. Sorry about the typo.